# Integrated Analysis of Gut Microbiome and Adipose Transcriptome Reveals Beneficial Effects of Resistant Dextrin from Wheat Starch on Insulin Resistance in Kunming Mice

**DOI:** 10.3390/biom14020186

**Published:** 2024-02-04

**Authors:** Xinyang Chen, Yinchen Hou, Aimei Liao, Long Pan, Shengru Yang, Yingying Liu, Jingjing Wang, Yingchun Xue, Mingyi Zhang, Zhitong Zhu, Jihong Huang

**Affiliations:** 1Food Laboratory of Zhongyuan, Luohe 462300, China; 2021930593@stu.haut.edu.cn (X.C.); 80891@hnuahe.edu.cn (Y.H.); aimeiliao@haut.edu.cn (A.L.); panlong@haut.edu.cn (L.P.); 2021920157@stu.haut.edu.cn (Y.L.); 2021930602@stu.haut.edu.cn (J.W.); 2021930597@stu.haut.edu.cn (Y.X.); 2022930656@stu.haut.edu.cn (M.Z.); zzt@stu.haut.edu.cn (Z.Z.); 2School of Biological Engineering, Henan University of Technology, Zhengzhou 450001, China; 3College of Food and Biological Engineering, Henan University of Animal Husbandry and Economy, Zhengzhou 450046, China; 81692@hnuahe.edu.cn; 4State Key Laboratory of Crop Stress Adaptation and Improvement, College of Agriculture, Henan University, Kaifeng 475004, China; 5School of Food and Pharmacy, Xuchang University, Xuchang 461000, China

**Keywords:** resistant dextrin, insulin resistance, inflammation, gut microbiota, transcriptome

## Abstract

Systemic chronic inflammation is recognized as a significant contributor to the development of obesity-related insulin resistance. Previous studies have revealed the physiological benefits of resistant dextrin (RD), including obesity reduction, lower fasting glucose levels, and anti-inflammation. The present study investigated the effects of RD intervention on insulin resistance (IR) in Kunming mice, expounding the mechanisms through the gut microbiome and transcriptome of white adipose. In this eight-week study, we investigated changes in tissue weight, glucose–lipid metabolism levels, serum inflammation levels, and lesions of epididymal white adipose tissue (eWAT) evaluated via Hematoxylin and Eosin (H&E) staining. Moreover, we analyzed the gut microbiota composition and transcriptome of eWAT to assess the potential protective effects of RD intervention. Compared with a high-fat, high-sugar diet (HFHSD) group, the RD intervention significantly enhanced glucose homeostasis (e.g., AUC-OGTT, HOMA-IR, *p* < 0.001), and reduced lipid metabolism (e.g., TG, LDL-C, *p* < 0.001) and serum inflammation levels (e.g., IL-1β, IL-6, *p* < 0.001). The RD intervention also led to changes in the gut microbiota composition, with an increase in the abundance of probiotics (e.g., *Parabacteroides*, *Faecalibaculum*, and *Muribaculum*, *p* < 0.05) and a decrease in harmful bacteria (*Colidextribacter*, *p* < 0.05). Moreover, the RD intervention had a noticeable effect on the gene transcription profile of eWAT, and KEGG enrichment analysis revealed that differential genes were enriched in PI3K/AKT, AMPK, in glucose-lipid metabolism, and in the regulation of lipolysis in adipocytes signaling pathways. The findings demonstrated that RD not only ameliorated IR, but also remodeled the gut microbiota and modified the transcriptome profile of eWAT.

## 1. Introduction

Insulin resistance (IR) plays a crucial role in metabolic syndrome (MetS) [1]. It typically manifests alongside hyperglycemia, hyperlipidemia, hypertension, and abdominal obesity, serving as a foundational factor in the pathogenesis of various metabolic diseases, including obesity, nonalcoholic fatty liver disease, type 2 diabetes mellitus, atherosclerosis and Alzheimer’s disease [2,3,4,5]. Consequently, alleviating IR has emerged as a pivotal strategy for early intervention in MetS. 

The three primary target organs for insulin metabolism are the liver, adipose tissue, and muscle [6]. White adipose tissue, a significant regulator of energy metabolism in the body, is particularly noteworthy. Inflammation within white adipose tissue is considered an initiating factor in the development of IR [7,8]. In adipose tissue, the main actions of insulin are to inhibit lipolysis by enhancing lipid synthesis and maintaining free fatty acid levels [9]. Furthermore, insulin facilitates glucose transport in adipose tissue through the PI3K/AKT/GLUT-4 pathway [10]. Previous studies have demonstrated that the byproducts of metabolism generated by gut microbiota can modulate adipose-tissue energy metabolism and insulin sensitivity [11,12]. 

The gut microbiota, a delicate ecosystem, plays a pivotal role in both health and disease [13]. Recent studies have shown a significant alteration in the composition of the gut microbiota in insulin-resistant patients, implicating the gut microbiota in carbohydrate and lipid metabolism as well as inflammatory responses, thereby mediating the development of metabolic diseases [3,14,15,16]. Moreover, numerous studies underscore the influence of diet on gut microbiota dysbiosis [17,18,19], and this behavior can be sustained. Therefore, it would be effective to alleviate IR through gut microbiota homeostasis. Dietary fiber, resistant to digestion and absorption in the small intestine, undergoes fermentation by gut microbiota, eliciting a specific response in the gut microbiota [20]. Meanwhile, dietary fiber exerts a profound influence on the concentration of bioactive compounds, such as short-chain fatty acids, bile acids, and branched-chain amino acids, originating from bacterial metabolites, which facilitate metabolic interactions between host and microbiota [21,22].

Metformin is widely recognized as a hypoglycemic agent, known for its ability to reduce blood glucose levels by inhibiting hepatic glucose production [23]. Recent studies have presented compelling evidence that metformin exerts notable effects on the gut microbiota, influencing both the small and large intestines [24,25]. In a rat model subjected to a high-fat diet, metformin demonstrated its capacity to modulate glucose-sensing pathways in the upper small intestine. This modulation was achieved through alterations in the gut microbiota, showing the intricate relationship between metformin, the gut microbiota, and the regulation of glucose-sensing pathways [26]. Furthermore, in a mouse model concentrating on colorectal cancer, metformin exhibited its ability to influence the relative abundance of genera associated with colorectal cancer. In particular, it played a role in inhibiting the formation of colorectal tumors induced by *Clostridium nucleatum* [27]. This underscores the fact that metformin modulates the microbiota and plays multiple roles. However, administering metformin in conjunction with a high-fat, high-sugar diet or at the onset of abnormal glucose metabolism is suboptimal [28], as it may result in gastrointestinal distress. Therefore, investigating functional foods as an alternative approach would be a more prudent and sensible course of action.

RD is generally produced by starch dextrination using specific processes, including thermal-acid, dry heat, and microwave methods [29]. RD, a soluble dietary fiber, exhibits diverse nutritional functions, including the potential to improve obesity, type 2 diabetes, sleep disorders in obese patients, non-alcoholic fatty liver disease, and polycystic ovary syndrome [30,31,32,33,34]. Moreover, RDs act mainly by modifying the composition of the gut microbiota, including the proliferation of probiotics and the inhibition of harmful bacteria [32,35,36]. However, previous studies reveal that the influence of RD on the gut microbiota is inconsistent, and possibly correlated with factors such as raw material source, molecular weight and processing methods [19,35,37,38,39]. In addition, there are fewer studies on whether RD affects the transcriptome profile of adipose tissue, and the effects of RD from wheat starch on the gut microbiota and its potential involvement in improving IR remain unclear.

In this study, our objective was to investigate glucose–lipid metabolic parameters and gut microbial changes on HFHSD Kunming mice following the intervention of RD from wheat starch, and to explore its potential effects on eWAT. Specifically, we employed 16S rRNA sequencing of cecum contents to discern the effects of RD intervention on gut microbial communities. Additionally, we conducted RNA sequencing analysis of eWAT to identify significantly enriched metabolic pathways and the underlying molecular mechanisms. This study provides new insights into the potential of RD intervention for improving IR.

## 2. Materials and Methods

### 2.1. Materials

Wheat starch was purchased from Liangrun Food Co., Ltd. (Xinxiang, China). Thermostable α-amylase (40,000 U/g) and amyloglucosidase (100,000 U/g) were purchased from Macklin Biochemical Technology Co., Ltd. (Shanghai, China). Biochemical kits for TC, TG, LDL-C, HDL-C were purchased from Beijing Solarbio Science & Technology Co., Ltd. (Beijing, China). ELISA kits for INS, IL-1β, and IL-6 were procured from Meimian Industrial Co., Ltd. (Yancheng, China). Metformin Hydrochloride Tablets were obtained from Jingfeng Pharmaceutical Co., Ltd. (Beijing, China). Adipose tissue Fix Solution was obtained from Servicebio Biochemical Technology Co., Ltd. (Wuhan, China).

### 2.2. Animal Experimental Design

Kunming mice (male; aged 4 weeks; classed specific-pathogen free) were obtained from Huaxing Experimental Animal Co., Ltd. (Zhengzhou, China, SCXY(Yu) 2019-0002). The mice were raised under standard specific conditions, with a room temperature of 22–26 °C, relative humidity ranging between 40 and 60%, a 12 h artificial light cycle, and proper ventilation. After one week of acclimatization, the mice were divided into four groups based on body weight (Figure 1). The average body weights did not differ significantly between four groups (*n* = 10 per group): normal diet (ND), high-fat, high-sugar diet group (HFHSD), HFHSD with RD intervention group (HFHSD+RD), and HFHSD with metformin control group (HFHSD+MC). The basal feed (mouse growth and breeding feed) was purchased from Beijing Keao Xieli Feed Co., Ltd. The feed formula for the basic diet (ND) is corn, soybean meal, fish meal, flour, yeast meal, vegetable oil, salt, multivitamins and minerals. The high-fat and high-sugar diet, consisting of 10% lard, 20% sucrose, 2.5% cholesterol, 1% sodium cholate, and 66.5% basic diet, was prepared by Huaxing Experimental Animal Farm Co., Ltd. (Zhengzhou, China). RD, administered via gavage at a dose of 6.2 g/kg/day, is equivalent to the recommended adult (60 kg) intake of about 0.5 g/kg/day. Similarly, metformin was administered by gavage at a dose of 200 mg/kg/day, translating to an equivalent adult dose of 16.2 mg/kg/day. During the intervention, the mice had unrestricted access to food and water, and fasting blood glucose levels were assessed every two weeks. The intervention lasted until the 8th week. All mice were fasted overnight for 12 h. All mice were anesthetized with isoflurane, sacrificed, and blood and tissue samples were collected. Blood samples were incubated at room temperature (25–27 °C) for 1 h, and then centrifuged at 4 °C and 2500 rpm for 25 min to separate the serum. Simultaneously, the remaining samples were rapidly frozen in liquid nitrogen and stored in a freezer at −80 °C. Additionally, the general health and welfare of the mice were monitored throughout the entire study duration. All animal experimentation protocols were conducted in accordance with the guidelines of the EU Directive (2010/63/EU) on the treatment and usage of laboratory animals. All experiments were approved by the Henan University of Technology Ethics Committee.

### 2.3. Preparation of Resitant Dextrin (RD)

The combination of thermal acidolysis treatment and enzymatic methods was used for the preparation of RD [40]. Wheat starch was suspended with HCl solution of 0.075 mol^−1^·L (1:1.5, *w*/*v*). After stirring for 30 min, the precipitate was dried in an oven at 40 °C for 24 h. The dried mixture was heated (180 °C, 90 min) to obtain crude dextrin. The crude dextrin was dispersed (1:3, *w/v*), with pH adjusting to 6.0 using 0.1 M NaOH. Subsequently, thermostable α-amylase (1%, *w/w*) was added and stirred at 95 °C for 120 min, then with pH adjusting to 6.0 utilizing 0.1 M HCl, amyloglucosidase (0.5%, *v/w*) was added and stirred at 60 °C for 60 min. The enzymes were then inactivated by heating the mixture at 100 °C for 10 min. The mixture was added to 4-times the volume of liquid ethanol (95%). The sample was the left, and was dried to obtain the RD [29]. The total fiber content of RD was determined using GB/T 22224-2008 (2), which revealed a dietary fiber content of 82.69%.

### 2.4. Blood Glucose and HOMA-IR Tests

Fasting blood glucose (FBG) was measured after an overnight fast lasting 12 h, using a glucometer (Roche, Shanghai, China). Fasting insulin levels (FINS) were determined employing Elisa kits (Meimian, Yancheng, China). For the oral glucose tolerance test (OGTT), the mice, after fasting for 16 h, received a glucose load (20%, *m*/*m*) of 2 g/kg body weight. Blood glucose levels were assessed at specific time periods (0, 30, 60, 90, and 120 min). The area under the curve (AUC) of OGTT and HOMA-IR index were calculated based on previous methodologies [41], with the following formula: HOMA-IR = FINS × FBG/22.5.

### 2.5. Histology Examination 

Briefly, the eWAT were fixed and then embedded in paraffin. Subsequently, the tissue samples were transversely sectioned into slices measuring 4–5 µm in thickness. These slices were stained with H&E to evaluate the pathological status of the eWAT. Finally, the images were analyzed with the CaseViewer 2.4 (3DHISTECH, Budapest, Hungary).

### 2.6. RNA-Seq Analysis 

Total RNA extracted from eWAT was assessed for purity and integrity using both the Nanodrop 2000 (Thermo Fisher, Waltham, MA, USA) and the Agilent 2100 (Agilent, Santa Clara, CA, USA). We chose the number of samples per group based on the RNA QC results and chose to retain 2 samples per group, following comprehensive analysis. After library construction, preliminary quantification and insert size calculation of the libraries were performed. Subsequently, RNA was sequenced via the Illumina Novaseq 6000 (Illumina, San Diego, CA, USA). The low quality reads and adaptor sequences were trimmed with Trimmomatic. The analysis pipeline involved aligning paired-end clean reads to the reference genome using HISAT2 v2.1.0 and quantifying read counts per gene with featureCounts v2.0.3. Transcripts per million (TPM) were then calculated for each gene, based on its length. Differential expression analysis was conducted using the edgeR software (v3.40.2), with *p*-values adjusted using the Benjamini & Hochberg method. Significantly differentially expressed genes were identified based on corrected *p*-values and absolute |log_2_FC| thresholds (|log_2_FC| ≥ 2, FDR < 0.01). Subsequently, clusterProfiler R software (v4.6.2) was utilized for statistical enrichment analysis of these genes in KEGG pathways. This service was conducted by Chi-Biotech Co., Ltd. (Shenzhen, China).

### 2.7. Gut Microbiome 16S rRNA Sequencing Analysis 

Total genomic DNA of the gut microbiome was extracted from cecum content samples using previously described methods [24]. The hypervariable region V3-V4 of the bacterial 16S rRNA gene was amplified with primer pairs 338F (5′-ACTCCTACGGGAGGCAGCAG-3′) and 806R (5′-GGACTACHVGGGTWTCTAAT-3′), employing an ABI GeneAmp^®^ 9700 PCR thermocycler (ABI, Foster, CA, USA). The PCR amplification of 16S rRNA gene was performed as follows: initial denaturation at 95 °C for 3 min, followed by 30 cycles of denaturing at 95 °C for 30 s, annealing at 55 °C for 30 s and extension at 72 °C for 45 s, single extension at 72 °C for 10 min, and end at 4 °C. The PCR product was extracted from 2% agarose gel, purified with the AxyPrep DNA Gel Extraction Kit (Axygen Biosci-ences, Union City, CA, USA), following the manufacturer’s instructions, and quantified using the Quantus™ Fluorometer (Promega, Madison, WI, USA). Purified amplicons were pooled equimolarly and subjected to paired-end sequencing on an Illumina MiSeq PE300 platform (Illumina, San Diego, CA, USA). Operational taxonomic units (OTUs) with 97% similarity cutoff were clustered using UPARSE version 7.1. This service was performed by Majorbio Co., Ltd. (Shanghai, China).

### 2.8. Statistical Analysis

All experimental results were presented as mean ± SDs. Multiple comparisons were conducted using one-way ANOVA with Tukey’s *post hoc* test. GraphPad Prism 7.0 (GraphPad Inc., San Diego, CA, USA) was used for the statistical analyses. A *p*-value < 0.05 was regarded as statistically significant. 

## 3. Results and Discussion

### 3.1. Resistant Dextrin Improves Metabolic Disturbance in HFHSD Kunming Mice

To investigate the effects of RD intervention on mice with glucose–lipid metabolism disorders induced by HFHSD, we examined various parameters including eWAT weight, glucose–lipid metabolism indicators, and serum inflammatory factor levels. By the eighth week of treatment, FBG, HOMA-IR index, AUC-OGTT, TC, and LDL-C in HFHSD mice were significantly elevated compared to those in ND mice (*p* < 0.01, one-way ANOVA with Tukey’s *post hoc* test). These data confirmed the successful establishment of the IR model in Kunming mice in this study. In contrast, RD intervention led to a significant reduction in FBG, HOMA-IR index, AUC-OGTT, TG, and LDL-C in HFHSD-fed mice (*p* < 0.001, one-way ANOVA with Tukey’s *post hoc* test, Figure 2A,C,D,F,H). The metformin intervention exhibited a similar effect. Changes in body weight (Appendix A), diet intake (Appendix A), FBG of OGTT-0min (Appendix A), and FINS (Appendix A) of Kunming mice during the intervention are shown in the Appendix A.

Compared to the HFHSD group, RD or metformin interventions resulted in a significant reduction in eWAT weight (*p* < 0.01, one-way ANOVA with Tukey’s *post hoc* test, Figure 3A). And the levels of serum IL-1β and IL-6 were elevated in HFHSD mice compared to those in ND, whereas they were significantly reduced in the HFHSD+RD and HFHSD+MC group (*p* < 0.001, one-way ANOVA with Tukey’s *post hoc* test, Figure 3C,D). H&E staining of eWAT showed cell enlargement in HFHSD mice, but RD or metformin intervention alleviated these effects (Figure 3B). The size of adipocytes could affect IR, and metabolic disorders have been associated with larger adipocytes [42]. This implies that RD possesses the capability to impact adipocyte size and potentially alleviate insulin resistance through specific pathways or actions.

### 3.2. Resistant Dextrin Modifes Transcriptome of EWAT in HFHSD Kunming Mice

Typically, white adipose tissue (WAT) is considered detrimental to health because of the presence of white adipocytes containing substantial lipid droplets. These cells play a role in influencing insulin sensitivity, lipid metabolism, and low-grade inflammation, ultimately contributing to the development of metabolic disorders [43]. However, the influence of RD on white adipose tissue remains uncertain. RNA-seq analysis of eWAT was conducted to further investigate the potential mechanisms underlying the improvement of IR in HFHSD-induced Kunming mice by RD intervention. Principal component analysis (PCA) of the transcriptome data revealed distinct clusters between the ND, HFHSD and HFHSD+RD groups (Figure 4A), indicating that mice were categorized based on dietary treatment and showed diet-specific transcriptional responses in all mice. In the comparison between ND and HFHSD groups, there were differentially expressed genes (DEGs) (|log_2_FC| ≥ 2, FDR < 0.01) (*n* = 1758), whereas HFHSD+RD versus HFHSD showed differences in more genes (*n* = 5364) (Figure 4B–D).

Through KEGG signaling pathway enrichment analysis, we observed a significant enrichment of the PI3K/AKT and AGE-RAGE signaling pathway in the diabetic complications, indicating substantial differences in gene expression associated with glucose metabolism between the ND and HFHSD groups (Figure 4E). We identified enrichment of DEGs in metabolic pathways including PI3K/AKT, PPAR, AMPK, glycerolipid metabolism and regulation of lipolysis in adipocytes when comparing HFHSD+RD with HFHSD groups (Figure 4F). The PI3K/AKT signaling pathway was the most significantly enriched, and it was found that the genes on the PI3K/AKT signaling pathway were significantly different between the HFHSD and HFHSD+RD group (|log_2_FC| ≥ 2, FDR < 0.01). Moreover, the genes were up-regulated following the RD intervention (Table 1), suggesting an improvement in IR and maintenance of the normal level of glucose–lipid metabolism in the body. The gene expression of the insulin metabolism (PI3K/AKT) signaling pathway between the HFHSD+RD and the ND group is described in the Appendix A. Previous studies have reported that the PI3K/AKT signaling pathway was important for insulin in maintaining glucose–lipid metabolic homeostasis [44]. And when IR occurred in adipose tissue, the release of numerous free fatty acids into the circulation aggravated dyslipidemia and IR, mainly due to the blockade of insulin metabolism signaling pathways [45]. 

Meanwhile, another data analysis revealed a significant upregulation of genes associated with lipid metabolism, namely, *CIDEA*, *OTOP1*, *CMKLR1*, *C/EBPα*, *PPARγ* and *ADIPOQ* between the HFHSD+RD group and the HFHSD group (|log_2_FC| ≥ 2, FDR < 0.01). These gene expression levels are shown in the Appendix A. The activation of *OTOP1* is believed to yield metabolic benefits for obesity, contributing to the maintenance of adipose immune homeostasis in obese patients [46]. *CIDEA1*, identified as a key gene for white adipose tissue “browning” [47], suggests that RD activates pathways promoting “browning”, potentially improving IR in adipose tissue. Regarding *CMKLR1*, its close association with inflammatory responses is documented. However, these findings were somewhat counterintuitive. Some studies suggest that *CMKLR1* deficiency may exacerbate glucose homeostasis and IR [48,49]. Conversely, other studies indicate that *CMKLR1* overexpression in adipose tissue significantly ameliorated liver histopathological changes in non-alcoholic steatohepatitis in rats [50]. These findings suggest that adipose tissue, with its distinct and specific role among tissues, requires investigation of the enigmatic role of eWAT in IR, especially after the RD intervention. The Gene Set Enrichment Analysis (GSEA) among the ND, HFHSD, and HFHSD+RD groups is presented in the Appendix A. Additionally, a heat map depicting the gene expression on the enriched pathway is included.

### 3.3. Resistant Dextrin Alters Gut Microbiota Composition in HFHSD Kunming Mice 

To investigate the effects of RD on the gut microbiota, we conducted the 16s rRNA sequencing analysis of the cecal contents. The data showed 782 common OTUs among the four groups (Figure 5A). Principal coordinate analyses (PCoA) of the weighted unifrac for these mice demonstrated good dispersion at the OTU levels, based on the within-group difference test ANOSIM (Figure 5B). There was no significant difference in the alpha diversity indices for OTU levels in the gut microbiota between ND, HFHSD, and HFHSD+RD. However, the intervention with metformin significantly reduced the alpha diversity of the gut microbiota. Detailed data can be found in the Appendix A. At the phylum level, the HFHSD+RD group showed higher relative abundances of *Bacteroidota*, *Proteobacteria*, and *unclassified_k_norank_d_Bacteria* compared to the HFHSD group. Conversely, *Desulfobacterota*, *Actinobacteriota*, *Patescibacteria*, *Verrucomicrobiota*, and *Deferribacterota* were decreased in the HFHSD+RD group (Figure 5C). This is further illustrated by circos plots of phylum-level flora differences between the four groups, and this is shown in the Appendix A. Interestingly, *Chloroflexi* was exclusively identified in the HFHSD+RD group, but absent in the other three groups, and it was considered a potentially anti-inflammatory bacterium [51]. At the genus level, the relative abundances of *Bacteroides*, *Faecalibaculum*, *Parabacteroides*, *norank_f_Ruminococcaceae*, *GCA-900066575*, *Alloprevotella*, *Muribaculum*, and *Parasutterella* were significantly increased in the HFHSD+RD group compared to the HFHSD group (Wilcoxon rank-sum test, *p* < 0.05, Figure 5D). RD intervention led to a reduction in the relative abundances of *unclassified_f_Lachnospiraceae*, *norank_f_Lachnospiraceae*, *Enterorhabdus*, *Candidatus_Saccharimonas*, and *Colidextribacter* (Wilcoxon rank-sum test, *p* < 0.05). Linear discriminant analysis effect size (LEfSe) was used to evaluate statistical significance and biologically relevant differences in the gut microbiota composition among the three groups of mice (Figure 5E). At the genus level, the LEfSe analyses revealed that *Bacteroides*, *Parabacteroides*, *Alistipes*, *GCA-900066575*, *norank_f_Ruminococcaceae*, and *Muribaculum* were highly enriched in the HFHSD+RD group compared to the HFHSD group. On the other hand, *Lactobacillus*, *Bifidobacterium*, *Enterorhabdus*, *unclassified_f_Lachnospiraceae* and *unclassified_f_Erysipelotrichaceae* were enriched in the HFHSD+MC group. It is worth noting that the HFHSD group showed an enrichment of *Colidextribacter*, which is typically considered to be a harmful bacterium. 

RD intervention modulates the homeostasis of the microbiota and acts by altering the structure of the gut microbiota, which is consistent with previous reports [34]. However, the specific changes in gut microbiota may vary, possibly due to differences in the methods of preparation or the raw materials of RD [52].The abundances of these prominent bacteria enriched by either the RD or metformin-supplemented groups have been associated with the improvement in glucose–lipid metabolism disorders, as well as the well-known ability of *Bacteroides*, *Muribaculum*, and *Bifidobacterium* to produce short-chain fatty acids. Some articles suggest that metformin has a limited impact on the regulation of gut microbiota in the cecum of high-fat diet-induced diabetic mice, primarily influencing the small intestine’s flora composition [24]. However, numerous studies increasingly emphasize the role of cecum flora, which can directly impact metabolism [53,54,55]. There is a possibility that the role of gut microbiota varies at different stages of glucose and lipid metabolism abnormalities. Therefore, it is imperative that subsequent experiments focus on detecting alterations in gut microbiota during high-fat and high-sugar dietary conditions, with particular emphasis on investigating the correlation between the changes and factors influencing disturbed glucose metabolism. 

### 3.4. Correlation of Gut Microbiota with Parameters of Glucose–Lipid Metabolism

The relationships between eWAT weight, serum lipid metabolism parameters (TC, TG, HDL-C, LDL-C), inflammatory cytokines (IL-1β, IL-6), glucose metabolism indicators (AUC-OGTT, HOMA-IR), and gut microbiota composition were examined using Spearman’s rank correlation analysis. Our results showed positive correlations between *Lachnoclostridium*, *Candidatus_Saccharimonas*, *norank_f__Oscillospiraceae* and *Colidextribacter* with markers related to glucose–lipid metabolism. Interestingly, the abundances of these bacteria exhibited a significant decrease following RD intervention, suggesting an improvement in glucose–lipid metabolism indicators. On the other hand, *Faecalibaculum*, *Bacteroides*, *Parabacteroides*, and *norank_f_Ruminococcaceae* showed negative correlations with glucose–lipid metabolism and inflammatory markers (Figure 6). In conclusion, these findings suggested that the RD intervention effectively abrogated the glucose–lipid metabolism disorders in HFHSD Kunming mice. This was evidenced by a decrease in TC, IL-1β, IL-6, LDL-C, HOMA-IR, AUC-OGTT and eWAT weight, accompanied by an increase in HDL-C. These improvements were primarily attributed to alterations in gut microbiota composition.

## 4. General Discussion

The importance of IR in the development of MetS is evident, given its role in hastening the onset of diseases like obesity and type 2 diabetes [56]. Previous studies have shown that RD improves glucose homeostasis and lipid parameters, modifies the composition of the gut microbiota, and ameliorates IR via liver signaling pathways [34,36], but they have tended to ignore the role of white adipose tissue. In this study, we assessed the effects of RD and metformin on HFHSD Kunming mice. Analysis of TC, TG, HDL-C, LDL-C, FBG, AUC-OGTT, HOMA-IR, IL-1β, and IL-6 revealed the effectiveness of both RD and metformin in improving IR and maintaining glucose–lipid metabolism homeostasis. Additionally, RD intervention resulted in a significant reduction in eWAT weight and prevented further enlargement of adipose cells. These beneficial effects suggest a positive influence on the early development of MetS.

Short-chain fatty acids are primarily produced by gut microbiota through the fermentation of dietary fiber. They play beneficial roles, including reducing obesity, lowering blood glucose, and improving IR [57,58]. The abundances of short-chain fatty acid producers, namely *Bacteroides*, *Muribaculum*, *norank_f__Ruminococcaceae*, and *Parabacteroides*, increased in the HFHSD+RD group. Notably, RD intervention emerged as a significant contributor to preventing IR by mediating the gut microbiota, and the relative abundances of *Bacteroides*, *Faecalibaculum*, *Parabacteroides*, *norank_f_Ruminococcaceae*, *GCA-900066575*, *Alloprevotella* and *Muribaculum* were significantly increased. These results were consistent with previous studies identifying these bacterial genera as primarily associated with improving glucose–lipid metabolism disorders [59,60,61]. *Faecalibaculum*, classified as lactic acid-producing bacteria, is believed to exert anti-obesity effects through the production of lactic acid [60]. Furthermore, a noteworthy negative association was observed between *Bacteroides* and IR-linked carbohydrate metabolites [14]. *Lactobacillus*, *Bifidobacterium*, and *unclassified_f_Lachnospiraceae* were enriched in the HFHSD+MC group and are considered probiotics. These findings suggest that both RD and metformin may contribute to improving Kunming mice IR by altering gut microbiota composition. The distinct variations in the effects of RD and metformin interventions on gut microbiota are evident, and are likely attributable to the secondary action of metformin on the gut microbiota. 

Among the various tissues constituting the WAT in male mice, eWAT is particularly susceptible to obesity development [62]. Previous studies suggested that RD supplementation could potentially alleviate adipose tissue inflammation by reducing macrophage infiltration, modulating macrophage polarization, and inhibiting NF-κB signaling, specifically in the eWAT [63]. Macrophage hypertrophy and chronic inflammation in adipose tissue are important factors in developing systemic IR [64]. The PI3K/AKT pathway, a classical pathway critical for insulin metabolism, exhibited significantly upregulated genes, suggesting that the RD intervention activated the ISR/PI3K/AKT pathway, and provided a new rationale for RD to ameliorate IR. Additionally, the PPAR signaling pathway influenced gene expression related to energy metabolism, cell development and differentiation. Our sequencing data also showed significantly increased expression levels of genes, including *C/EBPα*, *PPARγ*, *GLUT4*, and *ADIPOQ* in both the ND and HFHSD+RD groups, in contrast to the HFHSD group. In the terminal differentiation stage, which is characterized by the cessation of growth in targeted preadipocytes, *C/EBPα* and *PPARγ* play a pivotal role in inducing and sustaining the expression of key adipogenic genes (*GLUT4*, *AP2*, and *ADIPOQ*), crucial for the formation of functional adipocytes [65]. These alterations appear closely connected to the gut microbiota composition, implying that metabolites from the gut microbiota may mediate the role of the gut-adipose axis. This study provides novel insights into the mechanism of RD as a means to enhance insulin sensitivity. Certainly, there are some limitations in our study. Notably, there were noteworthy observations during the modeling of mice on a high-fat, high-sugar diet, particularly concerning the differences between females and males in glucose and lipid metabolism [66,67]. It is crucial to emphasize that, despite exposure to an identical diet, males demonstrated a heightened susceptibility to metabolic disorders. Therefore, validating the effectiveness of dietary interventions on female mice is necessary. This will lead to a more comprehensive understanding of the variability of metabolic abnormalities.

Subsequently, we consider investigating potential microbial changes in eWAT and biomarkers of the gut-adipose axis following RD intervention, aiming to further elucidate the link between gut microbiota and alterations in the gene expression. We propose conducting animal experiments to retrospectively investigate glycolipid metabolic parameters, changes in lipid transcriptome profiles, and alterations in gut flora throughout the process. Employing gut flora metabolomics and lipidomics would aid in exploring the effects of high-fat and high-sugar diets on stage-specific glycolipid metabolism changes in healthy mice, elucidating linkages, and uncovering underlying mechanistic processes. To ensure comprehensive insights, variations in animal sexes and feed formulations, particularly varying fat content (e.g., 15%, 30%, 45%, 60%), should be incorporated to investigate generalized mechanisms.

## 5. Conclusions

In summary, carrying out RD intervention (6.2 g/kg/day) on HFHSD Kunming mice can lead to a reduction in FBG, TG, LDL-C, AUC-OGTT, HOMA-IR, IL-1β, IL-6, and eWAT weight. These findings demonstrate the effectiveness of RD in alleviating IR in HFHSD-induced mice by reshaping the composition of gut microbiota. Moreover, RD intervention alters the transcriptomic profiles of eWAT in HFHSD mice. Our study provides new insights on the mechanisms through which RD intervention exerts beneficial effects in alleviating glucose–lipid metabolism disorders in HFHSD Kunming mice. The central role of gut microbiota and its complex interactions with the adipose tissue should be emphasized. These findings necessitate additional research to validate their clinical significance and explore potential therapeutic applications.

## Figures and Tables

**Figure 1 biomolecules-14-00186-f001:**
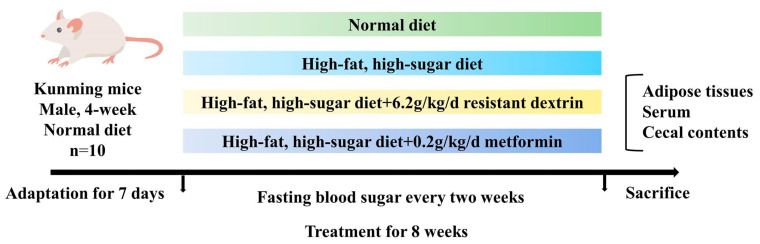
The animal experimental design.

**Figure 2 biomolecules-14-00186-f002:**
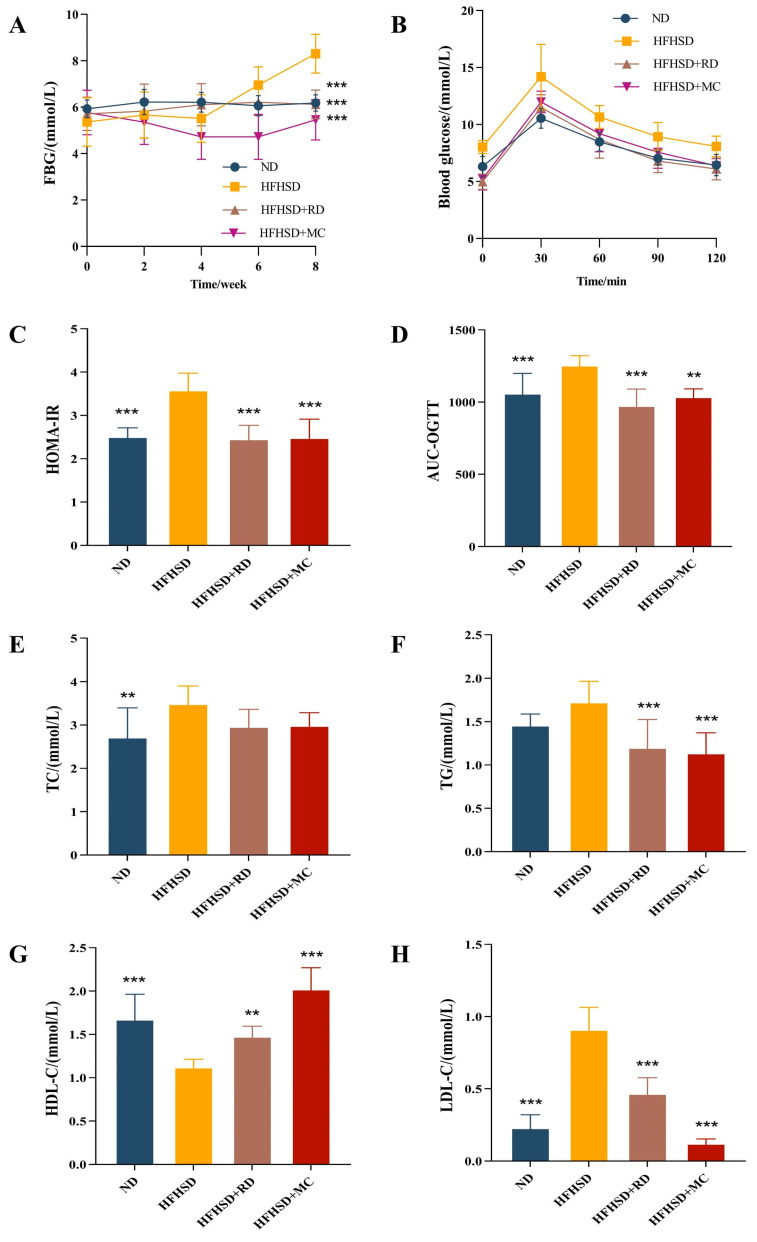
RD ameliorates glucose–lipid metabolism disorders in HFHSD Kunming mice (*n* = 10). (**A**) FBG every two weeks. (**B**) OGTT at the end of intervention. (**C**) HOMA-IR at the end of intervention. (**D**) Area under the curve (AUC) of OGTT. (**E**–**H**) Serum lipid profiles. ** *p* < 0.01, and *** *p* < 0.001 vs. HFHSD group.

**Figure 3 biomolecules-14-00186-f003:**
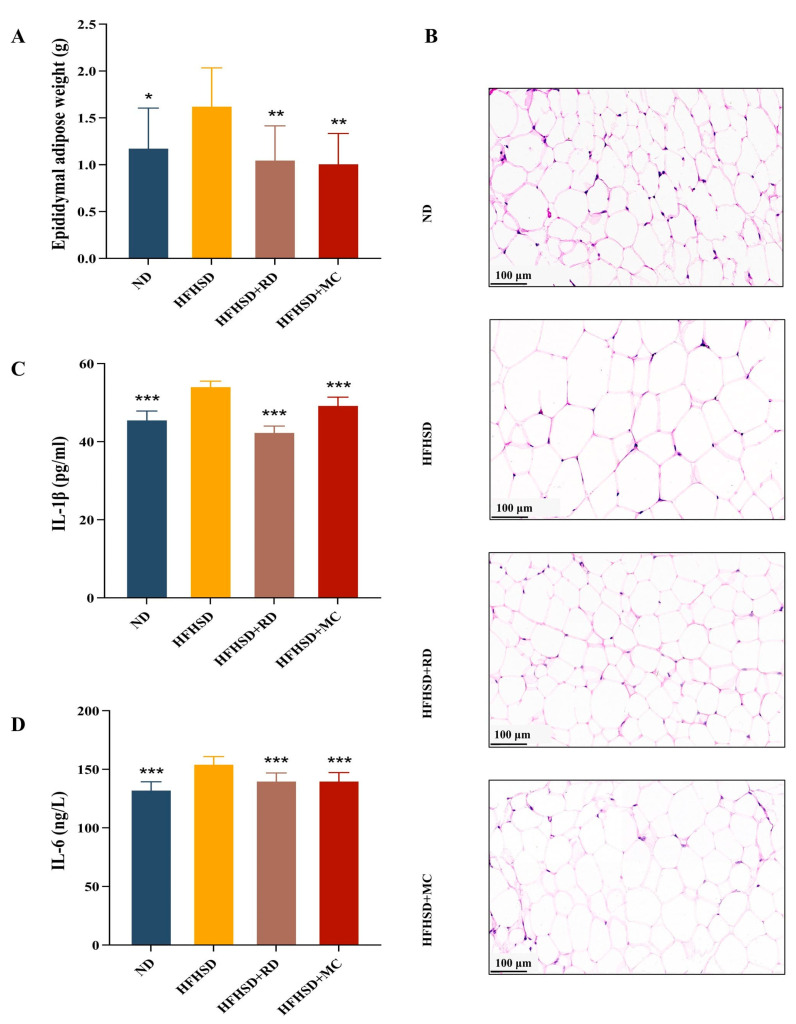
RD ameliorates eWAT weight, cellular hypertrophy and serum inflammation levels in HFHSD Kunming mice (*n* = 10). (**A**) The eWAT weight. (**B**) H&E staining. (**C**,**D**) Serum IL-1β and IL-6 levels. * *p* < 0.05, ** *p* < 0.01 and *** *p* < 0.001 vs. HFHSD group.

**Figure 4 biomolecules-14-00186-f004:**
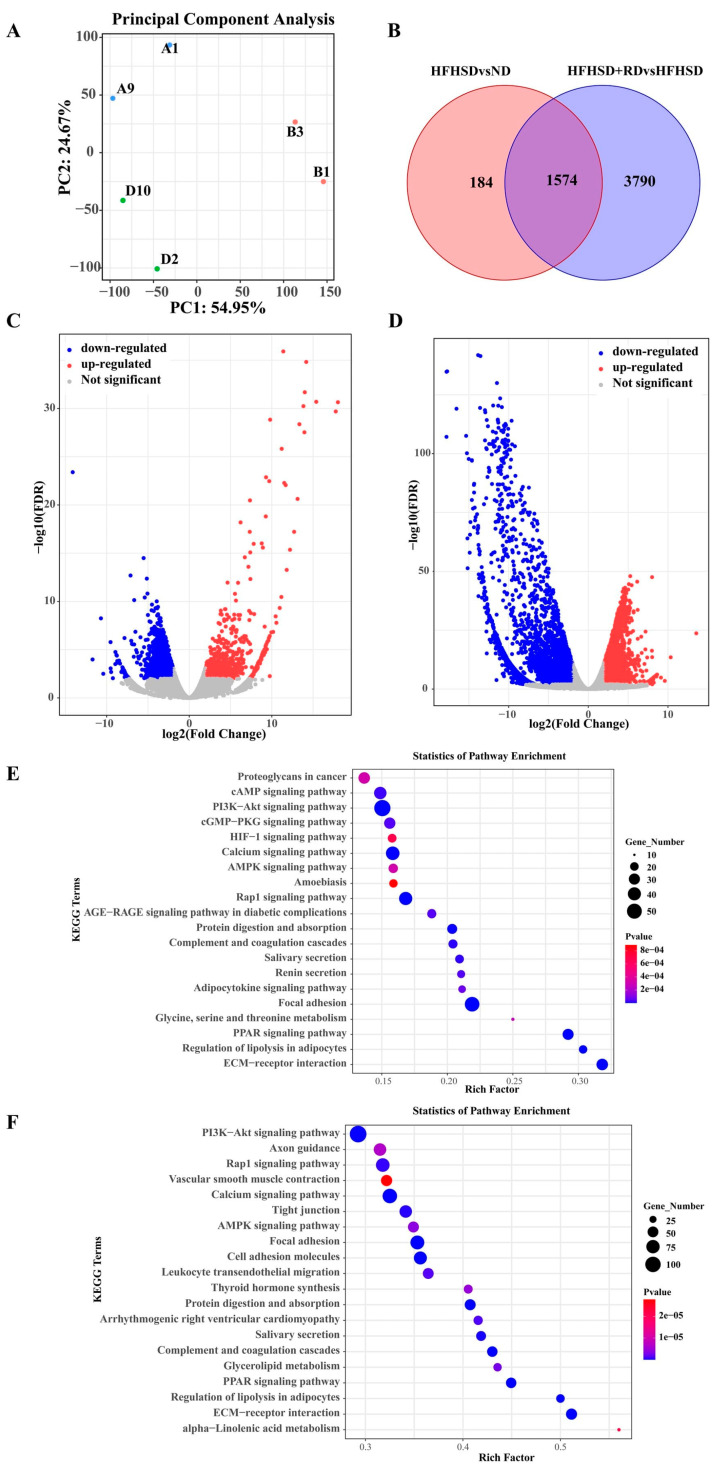
RD intervention modifies the eWAT transcriptome profiles in HFHSD Kunming mice (*n* = 2). (**A**) Principal component analysis, ND group: A1 and A9; HFHSD group: B1 and B3; HFHSD+RD group: D2 and D10. (**B**) Venn diagram showing the total number of DEGs in the HFHSD vs. ND and HFHSD+RD vs. HFHSD comparisons. (**C**) Volcano plot showing the significant DEGs for the HFHSD and ND groups. (**D**) Volcano plot showing the significant DEGs for HFHSD+RD and HFHSD groups. (**E**) KEGG enrichment analysis showing the top 20 signaling pathways affected by DEGs between HFHSD and ND groups (FDR < 0.01 |log_2_FC| ≥ 2). (**F**) KEGG enrichment analysis showing the top 20 signaling pathways affected by DEGs between HFHSD+RD and HFHSD groups (FDR < 0.01 |log_2_FC| ≥ 2).

**Figure 5 biomolecules-14-00186-f005:**
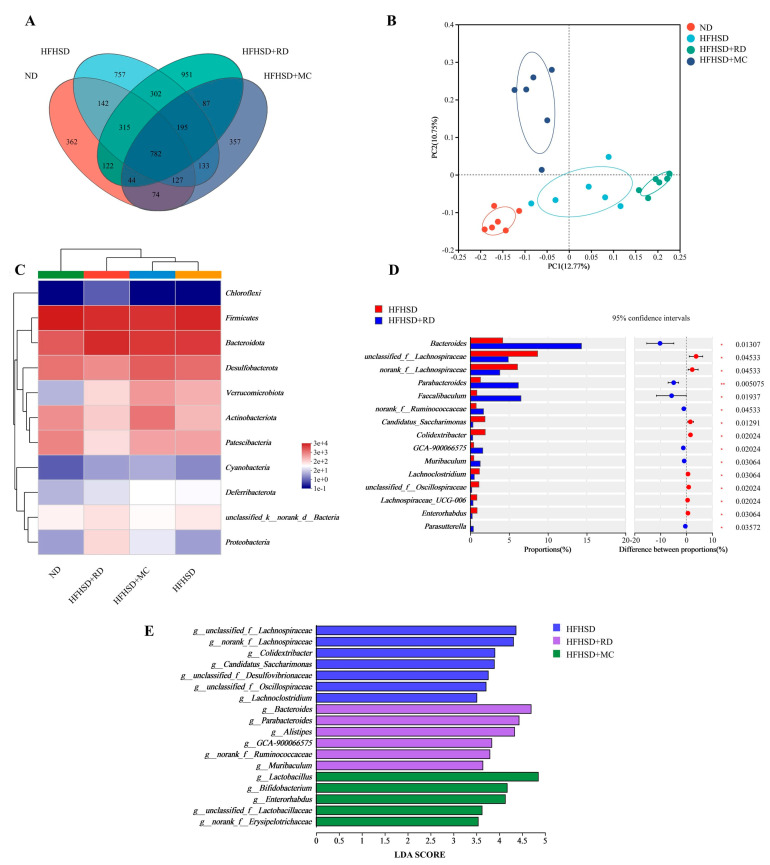
RD alters gut microbiota composition in HFHSD Kunming mice (*n* = 6). (**A**) Venn diagram depicting the number of bacteria at the OTU level between ND, HFHSD, HFHSD+RD, and HFHSD+MC groups. (**B**) Principal coordinate analyses based on OTUs of the microbial community in the four groups. (**C**) Gut microbiota composition at the genus level between the four groups of mice (abundance > 20). (**D**) Variances in the genus-level flora between HFHSD and HFHSD+RD mice employing the Wilcoxon rank-sum test. (**E**) LDA scores of differentially abundant taxa among HFHSD, HFHSD+RD, and HFHSD+MC mice, using the LEfSE method (LDA score > 3.5).

**Figure 6 biomolecules-14-00186-f006:**
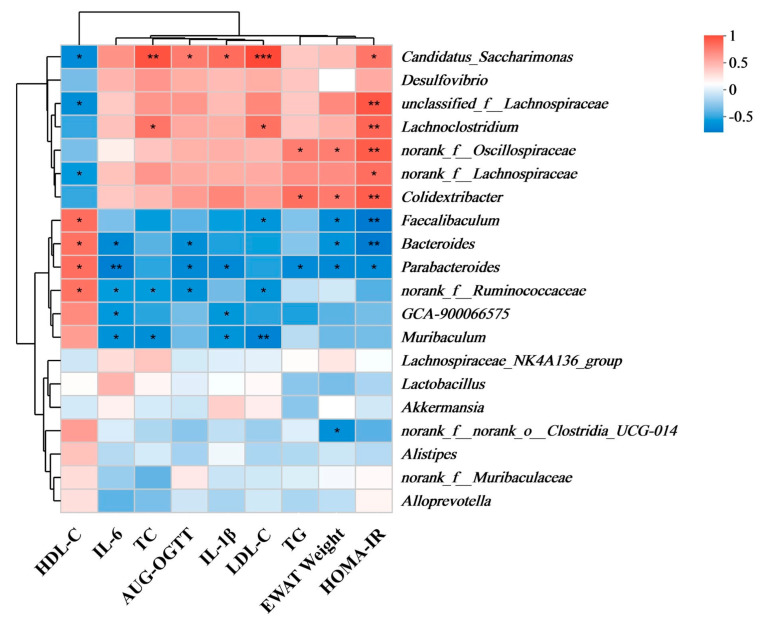
Correlation analysis of gut microbiota and glucose–lipid metabolism factors. (*n* = 6, * *p* < 0.05, ** *p* < 0.01, *** *p* < 0.001 indicates a significant difference).

**Table 1 biomolecules-14-00186-t001:** Gene expression of insulin metabolism (PI3K/AKT) signaling pathway.

Symbol	logFC	FDR	HFHSD	HFHSD+RD	Phenotype
Pik3r1	2.43	1.58 × 10^−10^	13.55	101.23	Disruptions can lead to impaired insulin signaling.
Pik3cb	2.26	2.34 × 10^−12^	6.48	42.90	Impaired PI3K activity may affect downstream signaling.
Pik3r6	2.20	2.49 × 10^−5^	1.33	8.29	Plays a role in insulin signaling and may impact glucose metabolism.
Akt2	2.69	1.42 × 10^−18^	28.54	251.00	Akt2 dysregulation is associated with insulin resistance.
Irs1	4.54	5.80 × 10^−20^	0.54	17.00	Impaired IRS1 can lead to insulin resistance.
Igf1	4.35	1.14 × 10^−39^	3.97	111.66	Deficiency levels can impact metabolism.
Slc2a4	3.78	5.19 × 10^−34^	11.18	207.59	Dysregulation can lead to impaired glucose uptake and insulin resistance.
Foxo1	2.17	1.16 × 10^−13^	4.57	28.26	Activation can enhance energy balance.

## Data Availability

The datasets generated for this study are available on request to the corresponding author.

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
