# Peer review of "Integrated Analysis of Gut Microbiome and Adipose Transcriptome Reveals Beneficial Effects of Resistant Dextrin from Wheat Starch on Insulin Resistance in Kunming Mice"

_biomolecules, 2024, doi:10.3390/biom14020186_

Round 1

Reviewer 1 Report

Comments and Suggestions for Authors

In the manuscript entitled: “Integrated Analysis of Gut Microbiome and Adipose Transcriptome Reveals Beneficial Effects of Resistant Dextrin on Insulin Resistance in Kunming Mice”, X. Chen et al aim to investigate and demonstrate the impact of a soluble dietary fiber, resistant dextrin (RD) from wheat, on insulin resistance through changes on the microbiota composition and in adipose tissue in high fat diet mouse model.

If the topic of the article is of interest, there are several concerned about the method and the interpretation of the results.

Major points

1/ Introduction

- The metformin and its medical uses and effects should be further described. 

- Why comparing the RD and Metformin action, what are the advantages or the inconvenient of using RD versus metformin? 

- It should be mentioned that the experiments were performed in mice

2/ The experimental design

- The experimental design includes only Kunming male mice and no female mice, while the authors generalize their conclusions and “recommend using RD as dietary supplements to alleviate glycolipid metabolism disorders. L.327

There are sex-specific responses to metabolic disorders in mouse as in human and therefore both sexes should be considered before generalizing observations and conclusions. The choice of only using male mice in the study should be addressed.

-The experimental design includes the administration of RD or metformin associated to the HFD continually from the beginning to the end of the experiment. However, metformin is administered to diabetics or pre-diabetics patients to counteract IR development, not to healthy subjects. Maybe it would have been better to design an interventional experiment and to administer the RD and metformin treatment to pre-diabetic mice (after more than 8 weeks of HFD for example). 

How the authors motivate the design of their experiment?

3/ The metabolic parameters and the metabolic tests

- The body weights of the mice are missing; they should be shown.

- The calculation of the HOMA index requires the insulin measurement during OGTT, the insulin data should be presented in the main figure or in supplementary data, since alone these data are already a good indicator of the metabolic status of the animals.

- In figure 2B. is the fasting glucose (baseline at 0min) significantly different between HFHSD and HFHSD+RD or HFHSD+MC?

- Did the RD or metformin affect the food intake of the mice?

4/ The characterization of the adipose tissue and inflammation

The authors write “H&E staining of eWAT showed inflammation and cell enlargement in HFHSD mice, but RD or metformin intervention alleviated these effects (Figure 3B)” L.183 

L.187-188 “These results may be attributed to the catabolic utilization of RD by the gut microbiota and the its potential interference in the NF-κB pathway related to inflammation [36,37]

No such a conclusion should be drawn based on the presented data. 

- The authors show only an H&E staining of adipose tissue but no staining for inflammatory markers such as F4/80 (macrophage) has been performed to characterize crown-like structure for example. The authors should consider doing it and counting the macrophage infiltration, that would give an idea of the inflammatory status of the adipose tissue upon RD or Metformin treatment come to vehicle.

- The authors have measured the serum level of pro-inflammatory cytokines IL1b and IL6. It would be interesting to measure the level of anti-inflammatory cytokines too and the level of adipokines such as leptin and adiponectin that have been involved in the development of IR. 

-The circulating LPS measurement (gram-negative bacterial product) could be a good indicator of the overall inflammatory status and of the integrity of the intestinal barrier. And it would make the link between the microbiota changes and the modifications observed at the metabolic parameters and in the adipose tissue. In obese/HFD model, the increased circulating LPS is known to accompany the microbiota changes and to increase IR.

- Moreover, the expression of some pro- and anti-inflammatory genes as well as western blot for local inflammatory markers would be of interest to characterize the impact of RD compared to vehicle or Metformin on adipose tissue inflammation. The NF-KB pathway should be then investigating.

5/ The transcriptomic analysis of the adipose tissue

- The authors report the analysis of only n=2 samples for each condition, that is too low to draw any conclusion. Why only 2 samples were analysed? Can the authors comment on it?

Moreover, there are no description of analysis method for the RNA-seq data in term of packages, cutoffs, filtering, parameter settings... Which pipeline has been used? Alignment to which reference genome? What was the method for determining the DEGs? What was chosen to determine expression levels (counts, FPKM, TPM…?)?

The raw data and generated results should be added in supplementary information.

The analysis of the data is too superficially performed. The action of RD is not compared to Metformin: what are the differences and similitudes between RD and metformin on the adipose tissue transcriptome? Which pathways are up-regulated vs down-regulated, which specific genes are changed by RD compared to Metformin or vehicle (…etc)?

- The two supplementary data (files) are not described or cited in the manuscript. There is no legend available for these figures.

Maybe simple qPCR analysis using larger number of samples (more than n=3) for specific markers would have been more informative.

6/ The Microbiota analysis of the caecum content

Overall, the section requires a much clearer and more extensive explanation of the analysis pipelines as for the transcriptome analysis. 

- The raw data and the generated results should be added as supplementary files to show more information.How were the OTUs obtained? For all analyses, it is unclear whether multiple comparison adjustments were performed. 

- L. 204 the authors write that Bacteroidota shows higher relative abundance in HFHSD+RD compared to HFHSD group, however from the Figure 4C no such conclusions can be drawn. 

- It is unclear whether LEfSe was used as a differential abundance analysis method or biomarker predictionLEfSe analysis has been a very popular method, however, current recommendations are not to use it with microbiota on its own due to the high number of false positives returned. If one still wants to use it, a combination of methods should be implemented. Also, since it was created to be a biomarker discovery tool, only the most weighted and significant bacteria will be discovered, which does not give a full view of the changes in the composition. 

- In Figure 4B, the PCA plot was used to visualize the clustering of the variables. Why ND seems closer to HFHSD than to HFHSD+RD? While the authors conclude that RD as an impact on the microbiota? How are the beta and alpha diversity? 

- IFigure 4E, the authors do not explain the choice of a LDA cutoff of 3.5. What was the reason for such a cut-off?

- It would be interesting to try to correlate/associate the changes in the microbiota composition with the changes in the adipose tissue transcriptome as done in Reference 21 (Murga-Garrido et al., 2021, Microbiome). 

7/ The conclusion about the impact of RD on microbiota composition and the link to changes in adipose tissue

The authors report independent changes upon RD treatment on adipose tissue transcriptome, blood metabolic parameters and microbiota in male mice fed an HFHSD

- What are the causal links between the different observations? It is not clear.

- The authors present strong conclusions along the manuscript

L.245-249: “In conclusion, these findings suggested that the RD intervention effectively abrogated the glycolipid metabolism disorders induced by the high-fat, high-sugar diet. This was evidenced by a decrease in TC, IL-1β, IL-6, LDL-C, HOMA-IR, AUG-OGTT and eWAT weight, accompanied by an increase in HDL-C. These improvements were primarily attributed to alterations in gut microbiota composition.”

L. 325-328: “These findings suggested that both RD and metformin may act to improve IR by altering gut microbiota composition. Therefore, we recommend using RD as dietary supplements to alleviate glycolipid metabolism disorders.”

However, only male mice have been used for this study: What are the sex or the species differences (mouse versus human)? And the methodology for the analysis of the different set of data is unclear and not exhaustive. 

Minor points

L.97: “classed SPF”, please clarify that it is “specific-pathogen free”

L.104: Is the normal diet (ND) “the basal feed?

L.101-102: How were the mice divided “based on body weight”?

L. 108-109: 6g/kg/day does not seem to be equivalent to 30g/day for a human (unless the human weighs 5 kg), please could the authors clarify how they calculate the dosage?

The number of mice for each assay should be clearly stated in the material and method section

L.116: “Refrigerator” or do the authors mean “freezer”? 

L.140: How many hours were the mice fasted? 16 or 12 hours? 

L. 215-216 the sentence seems to be disconnected from the rest of the paragraph. 

L. 267-270: There is redundancy, since the authors already took significant DEGs to check the altered pathways. 

L.283-284: A reference may be missing after “… responses is documented”?

L.272-276: The tempus is not correct, and it is unclear what is from the authors’ work and what is from other sources. It needs to be clarified.

L.277: “Meticulous” may not be the most appropriate adjective to qualify the data analysis as it is currently presented. 

Figure 2: “AUG-OGTT” or “AUC-OGTT”. 

Figure 3: The scale on the picture should be more visible.

Figures 4 and 5: It is difficult to read the text of the figures

Comments on the Quality of English Language

The English text is sufficiently clear.

Author Response

Thank you for dedicating your valuable time to review the manuscript. Your insights into the experimental design were both clear and innovative, and we collectively agreed that they were highly beneficial. We have incorporated your suggestions into the manuscript, providing additional explanations and relevant content. We sincerely appreciate your thoughtful feedback. Thank you once again.

Reviewer 2 Report

Comments and Suggestions for Authors

This study examined the negative effects of a high-fat and high-sugar diet on glucose-lipid metabolism, and the improvement effects of resistant dextrin in relation to the results of gut microbiota and transcriptome analysis of visceral fat cells. Interesting and important results have been obtained.

General comments

1, Since this research is results using mice, it is important to include a statement in abstract, introduction and conclusion that this research uses mice. Also, include a brief explanation of Kunming mice in introduction or method section because this strain may not be common.

2, The authors use the term 'glycolipid metabolism' in many parts, which is confusing because glycolipid is biological substance. Authors should reword this term, for example, to 'glucose-lipid metabolism'.

3, The prepared resistant dextrin is the most important factor in this research, so please describe its average molecular weight, molecular weight distribution, etc. as much as possible.

4, Metformin administration showed changes in intestinal bacteria that were different from RD. It is necessary to discuss how metformin, which does not reach the large intestine (cecum), affected intestinal bacteria.

5, Results section include many discussion, especially in microbiome and transcriptome parts. It seems to be better to change subtitles or rearrange Results and Discussion sections. “Results” to “Results and Discussion”, and “Discussion” to “General discussion” for example.

Individual comments

1, L 77: Better to change “wheat-resistant dextrin” to “resistant dextrin prepared from wheat starch”.

2, L 106: Show composition of “basal diet”.

3, L 107-109: It is necessary to explain how the RD dose of 6 g/kg/day is calculated to be 30 g/day in human,

4, L 113: It is necessary to describe what kind of anesthesia and how mice were killed.

5, L 139-140: Enter the name of the kit that measured insulin.

6, L.184-185, L 308-309, Fig. 3B: The authors describe the suppression of adipocyte hypertrophy by RD. Is there any data on quantifying the size of fat cells from the photos shown in Fig.3B?

7, L313-318: Is there any data showing that short chain fatty acid production increases with RD administration?

Author Response

Thank you very much for taking your valuable time to review my manuscript, and I appreciate your acknowledgment and insightful suggestions, with which I wholeheartedly agree. Following discussions with all the authors, I have implemented the revisions. Thank you for your unwavering efforts and acknowledgment.

Round 2

Reviewer 1 Report

Comments and Suggestions for Authors

Comment 1: Introduction

The authors answered to the comment and added the requested information.

Comment 2: The experimental design

The authors answered “Considering these findings and synthesizing results from past studies and our current investigations, it is evident that the physiological cycle of female animals, such as the estrous cycle, introduces more variability, potentially complicating the interpretation of experimental results.(…)”

“(…) our objective is to examine if a stage-specific high-fat diet among normal populations can alleviate the onset or delay the progression of glucose-lipid metabolism disorders through the consumption of functional foods, with a primary focus on nutritional aspects. Thirdly, we seek to ascertain if the high-fat diet phase among the normal population can pre-empt or postpone the onset of glucose-lipid metabolism disorders by integrating functional foods, concurrently structuring the experimental design with consideration for nutritional aspects.(…)”

Using both sex increases reproducibility and translationality. It is a misconception to consider that female models are more variable and therefore would “potentially” complicated interpretation of the presented study without evidence. From the statement of the authors, the aim of the study is to determine the impact of RD on “normal population” that includes both males and females. However, it can be argued that the presented data are only valid for male mice and should be taken carefully when it comes to extrapolate the real effect of the RD on a “normal population” since it is known that there are sex-differences in term of glucose, lipid metabolism and that for example, female mice tend to be more resistant to High fat diet induced metabolic disorders.

Furthermore, several anti-diabetic drugs have been developed and tested principally in males and have been shown in fact to have more side-effects or be less effective in women.

Tramunt B, Smati S, Grandgeorge N, Lenfant F, Arnal JF, Montagner A, Gourdy P. Sex differences in metabolic regulation and diabetes susceptibility. Diabetologia. 2020 Mar;63(3):453-461. doi: 10.1007/s00125-019-05040-3.

Arnetz L, Ekberg NR, Alvarsson M. Sex differences in type 2 diabetes: Focus on disease course and outcomes. Diabetes, Metab Syndr Obes. 2014; 7: 409-420. https://doi.org/10.2147/DMSO.S51301

Pettersson US, Waldén TB, Carlsson PO, Jansson L, Phillipson M. Female mice are protected against high-fat diet induced metabolic syndrome and increase the regulatory T cell population in adipose tissue. PLoS One. 2012;7(9):e46057. doi: 10.1371/journal.pone.0046057. Epub 2012 Sep 25. PMID: 23049932; PMCID: PMC3458106.

Oraha J, Enriquez RF, Herzog H, Lee NJ. Sex-specific changes in metabolism during the transition from chow to high-fat diet feeding are abolished in response to dieting in C57BL/6J mice. Int J Obes (Lond). 2022 Oct;46(10):1749-1758. doi: 10.1038/s41366-022-01174-4. 

The authors should add in the discussion the use of only male mice as a limitation of the presented study.

Comments 3: The metabolic parameters and the metabolic tests

Did the RD or metformin affect the food intake of the mice? 

The authors answered :“Compared to the HFHSD group, RD or metformin had a decreasing effect on food intake in mice, we also assessed their serum levels of the appetite-related hormone, leptin. Although some differences were noted, they did not reach statistical significance. We opted not to elaborate on this data in the paper, as our primary focus is on the effects of glucose and lipid metabolism. “

 The authors state that there is difference in food intake between the different diets/treatments. Please add the standard deviation and statistics on the graph.

Even a slight difference in food intake could impact the weight gain and the metabolism. How the decrease of food intake by the RD could be explained? Would it be possible that the observed data on metabolic parameters translate the impact of a decreased food intake more than the actual action of the RD.

The authors should address this point in the discussion.

Comment 4: The characterization of the adipose tissue and inflammation

The authors answered: “It is intriguing to investigate alterations in the inflammation levels of adipose tissue, along with the expression of inflammation-related genes and proteins influenced by resistant dextrin. This exploration is grounded in established serum biochemistry results, evaluating the levels of inflammatory factors in correlation with adipose tissue. Importantly, adipose tissue serves as the target organ in insulin metabolism. I will include the experimental design you described in my graduation thesis. “

“(…) the primary focus of this paper centers on examining the impact of wheat-resistant dextrin on glucose-lipid metabolism levels, the inflammatory state of adipose tissue, alterations in the composition of intestinal flora, and transcriptomic analysis of the eWAT in Kunming mice. (…)“

As the authors mention, they are interested in the “inflammatory state of the adipose tissue”. Thus, they have measured the plasma level of circulating cytokines and the levels indicate an overall inflammatory status under HFHSD and decreased overall inflammation with RD or Metformin, but it does not proof that the effect comes from the adipose tissue. 

If not doing IHC for immune cell infiltration the authors should extract from their RNA-seq analysis the expression of inflammatory genes to show that there is an impact of the RD on pro-inflammatory/anti-inflammatory markers not only showing KEEG pathways that do not give so much information about the inflammatory status of the adipose tissue as it is presented.

The author should add these data for this manuscript, not only for the graduation thesis.

The authors answered “In this article, we did not present the results of the measurement of four hormones: leptin, glucagon, dipeptidyl peptidase IV, and growth hormone-releasing peptide. “

Since the authors mention that they have measured the leptin level (see answer to comment 3), maybe it should be shown.

Comment 5: The transcriptomic analysis of the adipose tissue

The authors answer to the question about the number of replicates, they should mention it in their material and method description and add the FC and p-values cut-off description.

The authors compared the effect of Metformin to RD on HFHSD and investigate the different impact on the metabolic parameters and microbiota. It seems that if they have a comparable effect on the metabolic parameters, the two treatments impact differently the microbiota.

The authors wrote: ”(…)The distinct variations in the effects of RD and metformin interventions on gut microbiota are evident, likely attributed to the secondary action of metformin's effect on the gut microbiota.” L.393-394

So, one should expect differences also at the transcriptomic levels in the adipose tissue between Metformin and the RD. Are there different effects of the Metamorfin and the RD on the adipose tissue transcriptome?  Please, address this point.

And as an overall comment, the presentation of the transcriptom, including the results for Metformin+HFHSD should be presented before the microbiota.

The depository reference of the sequences should be in the material and method.  

L336-337

The authors wrote : “Meanwhile, data analysis revealed a significant upregulation of genes linked to lipid metabolism, namely, CIDEA1OTOP1CMKLR1C/EBPαPPARγ and ADIPOQ (|log2FC|2, FDR0.01).

The gene data described is not shown neither in a figure or a table.

The author should add either a figure or a table.

Comments 6: The Microbiota analysis of the caecum content

The authors answered : “(…) The current joint analysis of the transcriptome and the gut microbiome is exceptionally innovative, particularly concerning the gut microbial-fat axis, a relatively scarce area of study. (…)”

And the title of the manuscript is “Integrated Analysis of Gut Microbiome and Adipose Transcriptome Reveals Beneficial Effects of Resistant Dextrin from Wheat Starch on Insulin Resistance in Kunming Mice”

If so, the authors should try to correlate inflammatory markers (genes from transcriptomic analysis) of the adipose tissue and microbiota in order to integrate the different analysis.

Please, add the description of the microbiota analysis in the material and method section, with extraction method and data analysis pipeline.

The reference cited by the authors does not include any description of the analysis pipeline: 

Han J X, Tao Z H, Wang J L, et al. Microbiota-derived tryptophan catabolites mediate the chemopreventive effects of statins on colorectal cancer[J]. Nature Microbiology, 2023: 1-15. 

The depository reference of the sequences should be in the material and method.  

Comments 7: The conclusion about the impact of RD on microbiota composition and the link to changes in adipose tissue

The authors answered:  “(…) Concerning the relationship of the transcriptome, blood metabolic parameters, and the microbiota of mouse adipose tissue, our data, as mentioned in result section, indicate that the intervention of resistant dextrin led to changes in the intestinal bacterial flora of mice. Specifically, there was an increase in the abundance of short-chain fatty acid-producing bacteria, resulting in elevated levels of short-chain fatty acids in the intestinal cecum contents. This, in turn, impacted the glucose transport of intestinal epithelial cells and improved metabolic homeostasis of blood glucose. (…)”

The authors do not show any evidence that there is an impact on the intestinal glucose transport. Furthermore, if there are changes in short-chain fatty acid-producing bacteria, the authors do not show any data about the fatty acid composition in the caecum content (no metabolomic analysis). Their conclusions are therefore only extrapolation.

The authors should be more careful in the formulation of their conclusion and should clearly state when they do hypothesis.

The authors wrote: “(…) To contribute novel ideas, we must emphasize the unique role of white adipose tissue in glucose metabolism, rather than solely focusing on the liver's role. (…)”

One can say that there is change in adipose tissue transcriptome with RD treatment upon HFHSD. There are also some changes in the microbiota composition. And the metabolic parameters are modified. I agree that the study tends to show a rather beneficial impact of RD on these three domains in male Kunming mice. The authors should try in to integrating the microbiota data and the transcriptomic date (correlation between bacteria and gene markers for instance).

But still, there is a gap between these three observations and there is no evidence of link between, what is triggering what and how. 

At this point, the authors can only speculate, and it should be clear in the article.

Comments on the Quality of English Language

Some editing would be required but the English text is sufficiently clear. 

Author Response

We are deeply grateful for your invaluable support and guidance throughout my manuscript. Your professionalism, patience, and meticulous review of my work have been crucial in shaping the quality of my dissertation. Your insightful suggestions not only assisted in addressing these issues but also facilitated a more comprehensive understanding of my research methodology and conclusions. Furthermore, your comments have greatly augmented the academic rigor of my thesis, prompting me to further delve into my research questions and refine my scholarly approach.
